# Dietary Habits in Children with Respiratory Allergies: A Single-Center Polish Pilot Study

**DOI:** 10.3390/nu12051521

**Published:** 2020-05-23

**Authors:** Eliza Wasilewska, Sylwia Małgorzewicz, Marta Gruchała-Niedoszytko, Magdalena Skotnicka, Ewa Jassem

**Affiliations:** 1Department of Pulmonology and Allergology, Medical University of Gdansk, Debinki str 7, 90-211 Gdańsk, Germany; ejassem@gumed.edu.pl; 2Department of Clinical Nutrition, Medical University of Gdansk, Debinki str 7, 90-211 Gdańsk, Germany; sylwia.malgorzewicz@gumed.edu.pl (S.M.); marta.gruchala@gumed.edu.pl (M.G.-N.); 3Department of Food Commodity Science, Medical University of Gdansk, Debinki str 7, 90-211 Gdańsk, Germany; skotnicka@gumed.edu.pl

**Keywords:** nutritional status, obesity, dietary habits, allergy, pulmonary function, allergic rhinitis, asthma

## Abstract

Background: The rising trend in allergic diseases has developed in parallel with the increasing prevalence of obesity, suggesting a possible association. The links between eating habits and allergies have not been sufficiently clarified. Aim: To evaluate the nutritional status, eating habits, and risk factors of obesity and pulmonary function in children with allergic rhinitis. Materials and methods: We evaluated 106 children with allergic rhinitis (mean age 12.1 ± 3.4 years; M/F 60/46) from the Department of Allergology. Clinical data were collected regarding allergies, physical activity, nutritional status (Bodystat), dietary habits (Food Frequency Questionnaire validated for the Polish population), skin prick test with aeroallergens (Allergopharma), and spirometry (Jaeger). Results: All children suffered from allergic rhinitis; among them, 43 (40.6%) presented symptoms of asthma. There were differences between children with only allergic rhinitis (AR group) and children with both rhinitis and asthma (AA group) in pulmonary function (forced expiratory volume in one second (FEV_1_) 100 ± 11 vs. 92.1 ± 15.0; *p* < 0.05). A total of 84 children (79%) presented a normal body mass index (BMI) (10–97 percentile), 8 (7.5%) were underweight, and 14 (13.5%) were overweight or obese. There were no differences in body composition between the AR and AA groups. Incorrect eating habits were demonstrated by most of the children, e.g., consumption of three or fewer meals in a day (38%), sweets every day (44%), snacking between meals every day (80%), and eating meals less than 1 h before bedtime (47%). Compared to the AR group, the AA group was more likely to eat more meals a day (*p* = 0.04), snack more often (*p* = 0.04), and eat before sleeping (*p* = 0.005). Multiple regression analysis showed a significant association between high BMI and snacking between meals and low physical activity (adjusted R^2^ = 0.97; *p* < 0.05). Conclusions: The risk factors for obesity in children with allergies include snacking and low physical activity. Most children with respiratory allergies, especially those with asthma, reported incorrect eating habits such as snacking and eating before bedtime. A correlation between pulmonary function and body composition or dietary habits was not found.

## 1. Introduction

The incidence of allergic diseases in Poland is increasing concomitantly with improvements in living standards and the adoption of a Western lifestyle. The most common clinical manifestation of hypersensitivity to inhalant allergens is allergic rhinitis (AR), which is one of the strongest factors affecting the quality of life and contributing to missed or unproductive time at work and school. In Europe and the United States, a significant increase in allergic diseases has been observed in recent decades [1,2]. In addition, the multicenter, standardized, randomized Epidemiology of Allergic Disorders in Poland (ECAP) study showed a prevalence of AR among the Polish population of 36% based on self-reported nasal symptoms, and 29% as diagnosed by physicians [3].

Among children with allergies, decreased involvement in outdoor activities and increased problems with concentration, sleep problems, and headaches are seen; moreover, children with AR often also suffer from asthma [4]. It is estimated that up to 40% of people with AR also have asthma, and almost 70% of asthmatics present coexisting AR [5,6]. In the Polish population, the asthma rate was 8% in children and adolescents according to the ECAP study, of which 70% of asthmatics presented with AR, while asthma occurred in 40% of patients with AR [7].

It is known that not only hygiene habits and exposure to allergens, tobacco smoke, and environmental pollution, but also a poor-quality diet, high caloric intake, overweight, and obesity in children and adolescents are important environmental factors that are conducive to the development of allergies [8,9]. Epidemiological and clinical studies suggest a relationship between obesity and allergic rhinitis as well as bronchial asthma [10,11].

In recent years, a significant increase has been noted in the incidence of obesity in children and adolescents in many European countries [12]. Excess body mass was diagnosed in 2% of Polish children in the 1990s, and in 15% of children 20 years later [13,14,15]. The authors of these studies indicated increased changes in lifestyle and nutritional habits as the causes of increased childhood obesity, i.e., consumption of sweets and unhealthy food; limited consumption of fruits, vegetables, and whole grains; and limited physical activity. In recent decades, fast foods have become a significant component of the diet in Westernized high-income countries, and now also for young people in Poland.

Children with allergic diseases present numerous risk factors for poor nutrition status. There are few studies describing dietary habits and their impact on the nutritional status of people with respiratory allergies. Although allergies are chronic and common diseases, these issues have not yet been clarified. Moreover, early diagnosis of excess body weight in children with allergic diseases, including asthma, seems to be important due to the course and treatment of the disease [16]. Therefore, the aim of this work was to evaluate the pulmonary function, nutritional status, eating habits, and risk factors of obesity in children and adolescents with AR.

## 2. Methods

### 2.1. Study Design

In this single-center, cross-sectional study, we evaluated, for the first time pediatric patients with symptoms of persistent rhinitis who visited the Department of Allergology of the Medical University in Gdańsk, Poland, between 2015 and 2017. The study was performed in compliance with the Code of Ethics of the World Medical Association (Declaration of Helsinki). The study protocol was approved by the Gdańsk Medical University Ethics Committee, and written informed consent was obtained from the parents of each patient. The study was supported by local research grant no. ST-554.

### 2.2. Patients

Inclusion criteria for the study were as follows: (1) age 7–18 years old, (2) persistent allergic rhinitis (duration at least 6 months in the last 12 months) never diagnosed and never treated with antihistamine drugs, (3) ability to perform spirometry, and (4) signed consent from parents to participate in the study.

Patients were evaluated according to the study protocol by a multidisciplinary team (allergologist, pediatrician, dietician). Children with persistent rhinitis symptoms in the last 12 months who had never been diagnosed and treated with anti-allergic or anti-asthmatic drugs were enrolled in the study (Visit 1, screening). During the next visit (Visit 2), allergy was confirmed by skin prick test, AR was diagnosed according to Allergic Rhinitis and Its Impact on Asthma (ARIA) [6], and asthma according to Global Initiative for Asthma (GINA) guidelines [17]. The medical history and spirometry results indicated newly diagnosed asthma in 43 patients; therefore, patients were divided into two groups: allergic rhinitis (AR group), and allergic rhinitis and asthma (AA group). Anthropometry, bioimpedance assessment, and dietary habits based on Food Frequency Questionnaire (FFQ-6) were collected and compared between the two groups. The scheme of the study is presented in Figure 1.

Allergy background was confirmed with skin prick test to aeroallergens (*Dermatophagoides pteronyssinus*, *Dermatophagoides farinae*; cat, dog; *Alternaria alternata*, *Cladosporium herbarum*; pollens: grass mix, rye, birch pollen, alder, hazel; Allergopharma, Germany). Children with food allergies and atopic dermatitis were excluded from the study because of the frequent use of elimination diets.

Spirometry with a reversibility test (400 µg salbutamolum) was performed using a MasterScreen Pneumo spirometer, Jaeger Company, Germany. Forced expiratory volume in one second (FEV_1_), forced vital capacity (FVC) and forced expiratory flow (FEF_25–75_) were measured in accordance with the procedures recommended by the European Respiratory Society [18] and presented as percentage of predicted value (pv).

### 2.3. Nutritional Habits

Data were collected by face-to-face interviews using a researcher-designed standardized questionnaire based on the Food Frequency Questionnaire (FFQ-6) and validated for the Polish population [19]. The FFQ-6 is the most common dietary assessment tool used in large epidemiological studies of diet and health and is validated for the population. The self-administered FFQ-6 asks participants to report the frequency of consumption of approximately 62 line items over a defined period of time (last year). Each line item is defined by a series of foods or beverages. The FFQ-6 includes an assessment of eight food groups (sweets and snacks, dairy products and eggs, grain products, fats, fruits, vegetables and grains, meat products and fish, drinks). Respondents have a choice of six categories of food consumption frequency: (1) never or almost never, (2) once a month or less often, (3) several times a month, (4) several times a week, (5) daily, and (6) several times a day. The FFQ-6 also includes questions on eating habits, i.e., meal intake frequency and snacking between meals.

The following information was obtained: the number of meals in a day, amount of sweets consumed in a week, amount of fast-food eaten in a month, time of last meal before bedtime, and snacking between meals. Fast food was defined as mass-produced food prepared and served very quickly, with poor nutritional quality (hamburgers, takeaways, and carbonated soft drinks).

### 2.4. Physical Activity

The subjects were assigned to four categories depending on their level of physical activity: sedentary lifestyle (up to 2 h per week), low (3–5 h per week), moderate (6–7 h per week), and high (more than 8 h per week). One hour of physical activity corresponded to one hour of classroom attendance (45 min). Subjects were classified as having a sedentary lifestyle if only sometimes present during gym classes or not exercising at all. The low physical activity group attended gym class in school and an additional hour, e.g., swimming. Children identified as high activity trained in some kind of sport.

### 2.5. Nutritional Status

Body height was measured by stadiometer and body mass by electronic scale (Tanita Inc., Amsterdam, The Netherlands) by a nurse during the first visit. Body mass index (BMI) was calculated by dividing body mass in kilograms by the square of height in meters. Based on centile charts for sex and age for the Polish population—OLAF/OLA project—percentiles of BMI were specified [20]. According to the OLAF/OLA charts, above the 90th percentile is overweight, above the 97th percentile is obese, and below the 10th percentile is underweight. Body composition values of fat mass (FAT), fat-free mass (LEAN), and water content were measured via the bioimpedance method using a BodyStat 1500 (Bodystat Ltd., Ballafletcher, UK).

### 2.6. Statistical Analysis

Differences for somatic traits and between AR and AA groups were evaluated using Student’s t-test or using the Mann-Whitney test for asymmetrical distributions. Distributions of values for somatic traits were evaluated using the Kołmogorov-Smirnov test. Differences between qualitative data were compared using the χ^2^ test. The association between obesity risk factors and BMI percentile was determined using linear multivariate regression analysis. Differences were considered significant at *p* < 0.05. All analyses were carried out using the Statistica 10.0 software package.

## 3. Results

### 3.1. Patients

Allergic rhinitis was diagnosed in all 106 patients included in the study; among them, asthma was newly diagnosed in 43 (40.6%). Subjects with only allergic rhinitis were classified into the AR group and those with both allergic rhinitis and atopic asthma into the AA group (see Figure 1). All 106 children (100%) had a positive skin prick test to house dust mite (HDM; *D. farinae* and/or *D. pteronyssimus*), among which 40 (37.7%) were also positive to grass pollen (*n* = 29) and animals (*n* = 11).

Children in the AA group had a positive reversibility test and lower FEV_1_ % predicted volume than children in the AR group. The basic characteristics of the study groups and the spirometry parameters are presented in Table 1.

### 3.2. Eating Habits and Physical Activity

The results of the eating habits and physical activity assessment are presented in Table 2.

#### 3.2.1. Number of Meals

The majority of children reported eating three (33.4%), four (29.2%), or five (34.1%) meals per day. Children with AA ate more frequently than children with AR (χ^2^ = 12.9; *p* = 0.04).

Thirty-five children (34%) did not eat regularly; their meals were at different hours each day. There was no difference in meal regularity between AR and AA groups (χ^2^ = 0.26; *p* = 0.60).

#### 3.2.2. Sweets

Almost half of the patients (47; 44.3%) ate sweets every day, in comparison to 12 (11.5%) who consumed sweets only one day per week. There was no difference in the consumption of sweets between AR and AA groups (χ^2^ = 2.5; *p* = 0.88).

#### 3.2.3. Snacks

Eight-seven children (80%) snacked (sweet and salty snacks) between meals, 81.4% in the AA group and 79.3% in the AR group. There was no difference in snack consumption, but deeper analysis showed the AA group consumed more salty snacks than AR group (χ^2^ = 0.59; *p* = 0.04).

All overweight and obese children (AR and AA) snacked significantly more often between meals (χ^2^ = 9.46, *p* = 0.01) than children with normal BMI.

#### 3.2.4. Fast Food

Seventeen patients (16.0%) had never eaten fast food. Most of the children (84%) ate fast food; half of them (*n* = 52) ate it very rarely (once a month) and 8.7% ate it 4–6 times per month. There was no difference in fast food consumption between AR and AA groups (χ^2^ = 6.3; *p* = 0.50).

#### 3.2.5. Meals before Bedtime

It was found that children most often consumed their last meal of the day 0.5–2 h before bedtime; a total 12.4% (*n* = 13) did so 2 h before falling asleep, 19.8% (*n* = 21) did so much earlier (from 2.5 to 3 h before bedtime), and 49% (*n* = 52) ate the last meal <1 h before sleeping. Children with AA ate the last meal 1 h before sleep more frequently than those in the AR group (χ^2^ = 19.4; *p* = 0.005).

### 3.3. Physical Activity

The mean physical activity was 5 h per week. Most children (55%) reported 3–5 h/week physical activity. These children had only physical education (PE) at school and 1 h of additional activities after school (swimming or games). Children with AR reported low (63.4%; *n* = 40) and moderate (19.2%; *n* = 12) physical activity; similarly, children with AA reported low (42.0%; *n* = 8) and moderate (34.8%; *n* = 15) activity. There was no difference in physical activity between the AR and AA groups (χ^2^ = 13.1; *p* = 0.15). There was a negative correlation between physical activity level and BMI centile in the whole study population (Spearman’s R = –0.19; *p* < 0.05).

### 3.4. Nutritional Status and Body Composition

Obesity was diagnosed in six children (6.0%) and eight were overweight (7.5%). In the AA group, obesity was present in 4.7% compared to 6.9% in the AR group (χ^2^ = 3.58; *p* = 0.30). The results of body composition measurement are presented in Table 3. There was no difference between the AR and AA groups.

### 3.5. The Multifactorial Linear Regression Analysis

Multifactorial linear regression analysis showed an association (independent of age) between BMI percentile and both snacking and physical activity level (see Figure 2 and Table 4).

## 4. Discussion

In the present study, we evaluated the nutritional status and dietary habits of Caucasian children with allergic rhinitis alone or with co-existing asthma. Although all of the children presented respiratory allergy symptoms at least 12 months before the study, they were never diagnosed with allergies and had not previously been treated with an antihistamine or anti-asthmatic drugs before.

The most important finding of the study is that the majority of children with respiratory allergies reported incorrect eating habits and low physical activity, with 7.5% being overweight and 6.0% being obese. In the study population, excess body weight was significantly associated with snacking between meals and low physical activity.

### 4.1. Nutritional Status

Unexpectedly, the prevalence of overweight and obesity among allergic children was similar to the population of healthy children in Poland [13,14,21,22]. This aligned with data from the International Obesity Task Force (IOTF) showing that approximately 10% of children worldwide are overweight [12].

Although AR is a common disease, most authors focus on children with food allergies or asthma. These studies have suggested that adiposity indicators are associated with asthma, asthma severity, and atopy [23,24]. It is obvious that a positive energy balance is associated with changes in immune system functioning, including chronic inflammation, which is clearly an unfavorable phenomenon [22]. Overweight and obese children with allergic diseases have metabolic derangements, and obesity may have an impact on inflammation and clinical symptoms in asthma. The cause of impact of the obesity on asthma risk is still unknown. Potential etiologies include airway smooth muscle dysfunction from thoracic restriction, obesity-related circulating inflammation priming the lung, and obesity-related comorbidities mediating asthma symptom development. Studies suggest that obesity in children with asthma appears to be associated with greater airflow obstruction and a mildly diminished response to inhaled corticosteroids [25]. Additionally, anti-allergic and anti-asthmatic medications may be risk factors for obesity and physiological factors associated with puberty, also intensifying the tendency to gain weight in adolescents [23]. In our study, we did not take into account the effects of medicines because all of the children were newly diagnosed with respiratory allergies and had not been treated with an antihistamine or anti-asthmatic drugs. This may be one of the reasons for the relatively small number of children with obesity observed in our study.

Recent prospective evidence supports the notion that increased body weight precedes asthma development, but there is an ongoing debate as to whether obesity directly increases this risk or whether patients first experience asthma and then become overweight or obese, possibly because of respiratory constraints and reduced physical activity [26].

There are only a few studies on nutritional status and allergic rhinitis. A cross-sectional study of obesity indicators and AR in 8165 participants from the 2005–2006 National Health and Nutrition Examination Survey (NHANES) showed that overweight and obesity were associated with increased risk of AR in adults, but no such evidence was found among children [27].

Interestingly, in our study, although children with co-existing asthma were younger than and not as tall as the children with only AR, they had similar weight. However, there were no statistical differences between the number of overweight and obese children and body composition (FAT, LEAN) in the two groups. This is interesting because other authors have reported more than 50% of children with excessive body weight among children with asthma [28]. Spirometry parameters also did not correlate to BMI, body fat, and lean body mass content in the whole study group, although pulmonary function tests were lower in asthmatics. There were no differences in terms of family burden between allergy, asthma, obesity, exposure to tobacco smoke, and pet allergens.

### 4.2. Dietary Habits

Our study showed that incorrect eating habits were reported by most of the children with allergies, such as frequent consumption of fast foods and sweets, snacking between meals, and eating meals less than 1 h before bedtime.

Many studies have confirmed that fast-food consumption is linked to childhood obesity [29,30]. The multicenter International Study of Asthma and Allergies in Children (ISAAC) showed that fast food consumption is high in childhood (6–7 years), increases in adolescence (13–14 years), and is associated with higher BMI [31]. In our study, 35% of the children reported fast food consumption at least several times a month. This result is similar to the ISAAC results, showing that 27% of children and 52% of adolescents reported more than weekly fast food consumption [31]. We did not find an association between dietary habits and pulmonary function. There were also no differences in fast food consumption between children with asthma and those with only rhinitis. These results are different from those reported by other authors, suggesting that fast food consumption may contribute to the increasing prevalence of asthma, rhinoconjunctivitis, and eczema in adolescents and children [31]. Other results from case-control [32,33,34] and cross-sectional [35,36,37,38,39,40] studies indicate that consumption of fast foods is significantly related to current asthma and allergic rhinitis (pollen fever). Wang et al. suggested that the amount of processed foods eaten correlates with the frequency and severity of asthma [29].

Another important finding from this study was that approximately 80% of children with respiratory allergies snacked between meals every day. Moreover, although all children with excess body mass consumed more snacks compared to normal-weight patients and reported low physical activity, asthmatics consumed snacks more frequently (χ^2^ = 0.59; *p* = 0.04) and were more likely to eat their last meal of the day 1 h before sleeping (χ^2^ = 19.4; *p* = 0.001). Similar results were seen in the PANACEA study, which showed that among a population of 700 Greek children 10–12 years old with a 23.7% prevalence of asthma symptoms, almost half the children reported salty snack consumption ≥1 times/week [41]. In the cited study, consumption of salty snacks >3 times/week (vs. never/rarely) was associated with a 4.8 times higher likelihood of having asthma symptoms, irrespective of potential confounders. The authors noted that the association of salty snack eating and asthma symptoms was more prominent in children who watched television or played video games >2 h/day [41].

Unlike other researchers, we studied the times of meals consumed and found, interestingly, that almost half of the children ate in the last hour before bedtime. This incorrect habit was more common in children with asthma symptoms. There are well-known factors that affect and exacerbate inflammation in the lower respiratory tract in asthmatics, such as infection or gastroesophageal reflux. Eating immediately before bed might have contributed to the formation of gastroesophageal reflux and bronchial hyperreactivity in the studied group of children with AR. This is also interesting because children with AR differed compared to asthmatics in lung function (FEV_1_%pv), but not in nutritional status or other eating habits except for snacking and meals consumed less than 1 h before bedtime. Unfortunately, we did not study the symptoms of gastroesophageal reflux, and we, therefore, cannot form any specific conclusions.

Braithwaite et al. [31] postulated some possible mechanisms to explain the relationship between asthma and allergic disease and the consumption of fast food, which may involve higher concentrations of saturated fatty acids, trans fatty acids, sodium, carbohydrates, and sugar, as well as preservatives that may modulate immune reactions. Consumption of processed foods reduces the consumption of foods that are rich in protective nutrients, such as fruits and vegetables. A reduced intake of fruits and vegetables, which have antioxidative and anti-inflammatory properties, is likely to have an unfavorable impact on asthma prevalence/management [42]. Additionally, indications are that a diet poor in antioxidants is a key factor influencing the development of allergic diseases; a Western lifestyle and processed food consumption can also cause reduced exposure to microbial products and a changed microbiome, which are thus possible causes of the increase in allergic disease [43].

### 4.3. Physical Activity

In our study, children from both groups (AR and AA) in over 50% reported low physical activity. It is well known that a lack of exercise increases the risk of obesity. This was confirmed by research in the Phase 3 ISAAC trial, in which television viewing (5+ h/day vs. <1 h/day, *p* < 0.001) (the group with low physical activity) was statistically significantly associated with higher BMI in comparison to vigorous physical activity (3+ h/week vs. never, *p* < 0.001) (the group with high physical activity) in adolescents. The authors also suggested that current behaviors are more important than other factors such as birth weight, breastfeeding, current maternal or paternal smoking in early childhood in the development of obesity [24].

A few studies have shown a relationship between physical activity and allergy. In the ISAAC study, associations were found between vigorous physical activity and a sedentary lifestyle for 13-year-olds with allergic rhinoconjunctivitis. Mitchell et al. showed that several hours of TV viewing was associated with symptoms of current asthma in adolescents [24]. Similarly, studies indicate that physical activity could be protective against the development of asthma [44]. On the other hand, Byberg et al. found no association between physical activity and allergic rhinoconjunctivitis [45].

Our results do not contradict the association between pulmonary function and physical activity but show a correlation between physical activity level and BMI percentile in the whole study population (Spearman’s R = –0.19; *p* < 0.05).

Our study provides two very important issues in the study of respiratory affections such as allergy and asthma. These are the fact that this is the first study on nutrition carried out in newly diagnosed AR teenagers, before any medication that could mislead any result and the fact that this is another study from a few existent about nutrition in respiratory allergy. The limitation of our study is the relatively small group of patients; therefore, a more accurate analysis was not possible, for example based on age or sex. This study is not generalizable to the Polish population because it was performed in a clinical sample of children. Although the relationship between incorrect dietary habits, low physical activity, and obesity in children with respiratory allergies is supported by our findings, no conclusions about causality can be made due to the cross-sectional design.

Further studies with large groups are necessary to determine the relationship between respiratory allergy, body weight, and diet.

## 5. Conclusions

The risk factors of obesity in allergic children were found to be snacking and low physical activity. Most children with respiratory allergies, especially asthmatics, reported incorrect eating habits such as snacking and eating before bedtime. A correlation between pulmonary function and body composition or dietary habits was not found.

Our study also indicated that in groups of children with respiratory allergies, there is a need for correction of diet and lifestyle. We suggest that early dietary correction may be helpful for children with allergic rhinitis and a high risk of asthma.

## Figures and Tables

**Figure 1 nutrients-12-01521-f001:**
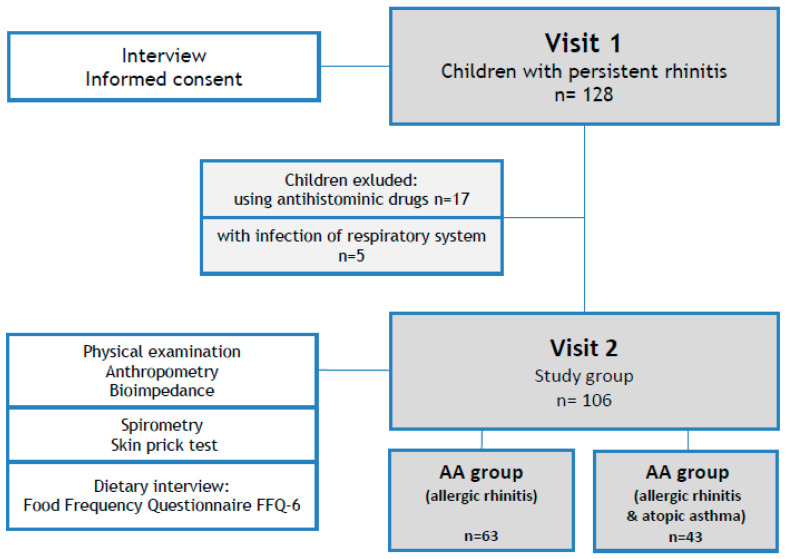
Study design.

**Figure 2 nutrients-12-01521-f002:**
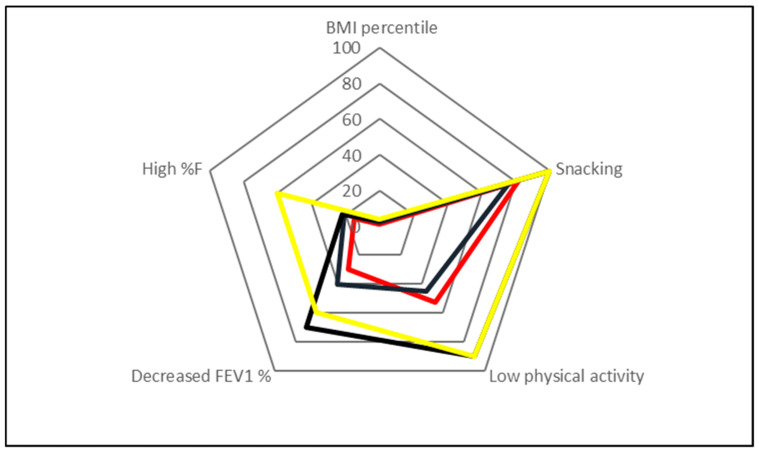
Association between body mass index (BMI) percentile and risk factors for obesity in all studied groups (adjusted R^2^ = 0.97; *p* < 0.05). %F: percentage of body fat; FEV_1_: forced expiratory volume in one second. Legend: red line means < 25th percentile, black line 25^th^–89th percentile, grey line 90^th^–97th percentile and yellow line >97th percentile.

**Table 1 nutrients-12-01521-t001:** Clinical and lung function characteristics of patients.

Parameters	All Children * *n* = 106	AR Group *n* = 63	AA Group *n* = 43	*p*-Value AR vs. AA
M/F	60/46	38/25	22/21	0.34
Age (years) mean ± SD (range)	12.2 ± 3.5 (7–18)	13.3 ± 3.5 (7–18)	11.5 ± 3.2 (7–18)	0.01
Tobacco smoking exposure *n* (%)	21 (19.8%)	15 (24%)	6 (28%)	0.21
Animal at home n (%)	49 (50.7%)	32 (50.7%)	17 (39.5%)	0.25
Family allergies n (%)	65 (61%)	34 (53%)	31 (72%)	0.09
Spirometry mean % pv ± SD				
FEV_1_	95.4 ± 16.3	100.0 ± 11.1	92.1 ± 15.0	0.05
FVC	95.1 ± 10.0	97.1 ± 10.1	94.0 ± 10.9	0.13
FEV_1_%FVC	108.8 ± 9.9	102.0 ± 4.4	99.0 ± 9.2	0.16
PEF	86.7 ± 16.0	90.2 ± 15.4	84.2 ± 15.9	0.06

AR: allergic rhinitis; AA: atopic asthma; FEV_1_: forced expiratory volume in one second; FVC: forced vital capacity; PEF: peak expiratory flow; pv: predicted value; n: number of subjects. * All children (*n* = 106) had a positive skin prick test.

**Table 2 nutrients-12-01521-t002:** Eating habits and physical activity assessment.

Parameters	All Patients *n* = 106	AR Group *n* = 63	AA Group *n* = 43
Meals (number per day)			
2	4 (3.7%)	3 (4.7%)	1 (2.3%)
3	35 (33.0%)	19 (30.1%)	16 (37.2%)
4	31 (29.2%)	16 (25.6%)	15 (34.8%)
5 or more	36 (34.1%)	25 (39.6%)	11 (25.7%)
Sweets (days per week)			
1	12 (11.5%)	7 (11.1%)	5 (11.6%)
2–3	29 (27.3%)	19 (30.1%)	10 (23.2%)
4–6	18 (16.9%)	10 (15.8%)	8 (18.6%)
every day	47 (44.3%)	27 (43.0%)	20 (46.6%)
Fast food (days per month)			
never	17 (16.0%)	12 (19.0%)	5 (11.6%)
1	52 (49.0%)	32 (50.7%)	20 (46.6%)
2–3	27 (25.4%)	13 (20.6%)	14 (32.5%)
4–6	9 (8.7%)	6 (9.7%)	3 (7.0%)
every day	1 (0.9%)	0 (0.0%)	1 (2.3%)
Last meal before sleep (hours to bedtime)			
<1	52 (49.0%)	30 (47.6%)	22 (51.1%)
1	20 (18.8%)	7 (11.1%)	13 (30.2%)
2	13 (12.4%)	11 (17.4%)	2 (4.6%)
>2	21 (19.8%)	15 (23.9%)	6 (14.1%)
Snacking between meals			
yes	85 (80%)	50 (79.3%)	35 (81.4%)
no	21 (20%)	13 (20.7%)	8 (18.6%)
Physical activity			
sedentary lifestyle	9 (8.4%)	4 (6.3%)	5 (11.6%)
low	58 (54.7%)	40 (63.4%)	18 (42.0%)
moderate	27 (25.4%)	12 (19.2%)	15 (34.8%)
high	12 (11.5%)	7 (11.1%)	5 (11.6%)

AR: allergic rhinitis; AA: atopic asthma; n: number of subjects.

**Table 3 nutrients-12-01521-t003:** Nutritional status and body composition in studied groups.

Parameters	All Patients	AR Group *n* = 63	AA Group *n* = 43	*p*-Value AR vs. AA
Anthropometric data, mean ± SD (range)				
Weight (kg)	47.1 ± 17.9 (18–98.2)	51.3 ± 17.9 (21–95)	44.5 ± 17 (18–92)	0.06
Height (cm)	154.6 ± 19.1 (110–185)	160.3 ± 17.1 (116–182)	151.4 ± 19.2 (110–185)	0.03
BMI (percentile)	45.5 ± 32.1 (1–99)	41.6 ± 31.1 (5–99)	47.9 ± 33.1 (1–99)	0.63
BMI, *n* (%)				
Underweight BMI <10th percentile	8 (7.5%)	5 (7.9%)	3 (6.9%)	0.81
Normal BMI, 10th–90th percentile	84 (79.0%)	52 (82.5%)	32 (74.4%)	0.43
Overweight BMI, 90th–97th percentile	8 (7.5%)	3 (6.9%)	5 (11.0%)	0.57
Obesity BMI, >97th percentile	6 (6.0%)	3 (4.7%)	3 (6.9%)	0.89
Body composition; mean ± SD (range)				
Body fat (%)	29.6 ± 20.6 (1–90)	20 ± 15.4 (3–70)	21.5 ± 17 (1–90)	0.14
Body fat (kg)	9.2 ± 4.8 (1–30)	8.2 ± 4.7 (2–31)	7.9 ± 3.9 (1–17)	0.94
LEAN (%)	70.3 ± 20.6 (8.9–89)	79 ± 16.5 (21–89)	78.1 ± 17.1 (8.9–56)	0.42
LEAN (kg)	27.9 ± 17.3 (2–24)	39.5 ± 17.6 (6–24)	36.8 ± 17.4 (2–24)	0.23

AR: allergic rhinitis; AA: atopic asthma; BMI: body mass index; LEAN: lean body mass.

**Table 4 nutrients-12-01521-t004:** Multivariate regression model predicting BMI value (adjusted R^2^ of the model was 0.97, *p* < 0.05).

Regression Model	B	Standard Error	Beta	*p*-Value
Constant	16.7	2.84		<0.001
Snacking	2.07	0.94	0.21	0.03
Fat%pv	−0.05	0.02	−0.21	0.058
FEV_1_%pv	0.03	0.02	0.11	0.23
Physical activity	–1.02	0.46	–0.21	0.028

FEV_1_: forced expiratory volume in one second; B: Regression coefficient B; Beta: beta standardized regression coefficient.

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
