# Peer review of "Dietary Habits in Children with Respiratory Allergies: A Single-Center Polish Pilot Study"

_nutrients, 2020, doi:10.3390/nu12051521_

Round 1
Reviewer 1 Report
Dear authors,
Thank for your having taken into account my comments and for the extensive revision.
Author Response
Thank you for accepting our corrections
Reviewer 2 Report
To the authors:
After the great effort and work put into the manuscript, now entitled “Dietary habits in children with respiratory allergies: a single-center Polish pilot study.” By Eliza Wasilewska, Sylwia Małgorzewicz, Marta Gruchała-Niedoszytko, Magdalena Skotnicka, Ewa Jassem, I must say authors have successfully addressed the comments from the previous revision.
This study provides two very important issues in the study of respiratory affections such as allergy and asthma. These are the fact that this is the first study on nutrition carried out in newly diagnosed AR teenagers, before any medication that could mislead any result, and the fact that this is another study from few existent about nutrition in respiratory allergy.
However, I found minor aspects that I consider overshadow the good work:
- On line 43, allergic rhinitis has been defined as AR, so I suggest that throughout the text make consistent and use this abbreviation, e.g. in line 74.
- Line 73. It’s a bit odd to see written after the introduction the “Aim”. Usually, this is included into the sentence, like, Therefore, the aim of this work was to evaluate…
- Line 79, change the “for the first time” of place to “we evaluated for the first time, pediatric patients with” in line 77. Otherwise, the phrase sounds strange.
- Figure 1 is in very low resolution.
- Line 90, please change “y” by “years old”.
- Lines 157-159. Something is not correct one % or an “n” is leftover. “All 106 children (100%) had a positive skin prick test to house dust mite (HDM; D. farinae and/or D. pteronyssimus), among which 41 (40.1%) were also positive to grass pollen (n = 29) and animals (n = 11).”
- Table 1. Remove the line with the “106 (100%)”, this can be included as a footnote.
- Table 1. Change the name to something more accurately such as “Clinical and lung function characteristics of patients.”
- Line 166, the title is eating habits but talks also about physical activity, so include it in the title.
- Line 172 and 193. “Children with asthma” were defined as AA, so use the definition.
- Line 192. “from 2.5 to 6 hours before bedtime”, please check this sentence, it is strange to think that last meal was 6 hours before sleep.
- Line 197, the definition PE has not been defined so write both words.
- Line 203. (same as before) in the title add that not only nutritional but also body composition will be shown.
- Line 205. Please review, I think this is incorrect: “obesity was present in 11.1% compared to 6.9% in the AR group”, while according to the table, obesity was 4.7 and 6.9%.
- Line 211. All the manuscript has been divided into small sections, and here the multifactorial linear regression analysis was just introduced. Please add a title.
- Line 217. Add to the figure 2 caption, the meaning of yellow, red, grey and black lines.
- Table 4, please add in the footnote the meaning of B and Beta
- Line 245, this is a formal text so I recommend you to avoid contractions such as “obesity’s”.
- Lines 327-330. This paragraph was fully copied from results. Avoid to duplicate the information, this will make to the reader to lose attention or credibility to the work. So eliminate or re-write in other form to introduce the discussion.
- Lines 331 to 334. This sentence is not understood. What are the 2 groups compared in the ISAAC trail phase 3.
- Line 335. “current behaviors are more important than other factors”, instead of write other factors, name them.
- Line 346. “The value of our study is its complex examination…”, I do not understand what the authors mean with this sentence, but I suggest to use any idea from the begging of my letter.
- Finally, sorry to be so insistent in this but for the scientific community, is important that p value, is in italics as p.
Author Response
Answer for Reviewer 2.
Thank you for accepting previous correction we have made. Also, we have included answer for the Reviewer's suggestions below.
- On line 43, allergic rhinitis has been defined as AR, so I suggest that throughout the text make consistent and use this abbreviation, e.g. in line 74.
We have done it.
- Line 73. It’s a bit odd to see written after the introduction the “Aim”. Usually, this is included into the sentence, like, Therefore, the aim of this work was to evaluate…
We changed to: Therefore, the aim of this work was to evaluate pulmonary function, nutritional status, eating habits, and risk factors of obesity in children and adolescents with AR.
- Line 79, change the “for the first time” of place to “we evaluated for the first time, pediatric patients with” in line 77. Otherwise, the phrase sounds strange.
We changed to: In this single-center, cross-sectional study, we evaluated, for the first time patients with symptoms of persistent rhinitis who visited the Department of Allergology of the Medical University in Gdańsk, Poland , between 2015 and 2017.
- Figure 1 is in very low resolution.
We added figure in proper resolution
- Line 90, please change “y” by “years old”.
We have done
- Lines 157-159. Something is not correct one % or an “n” is leftover. “All 106 children (100%) had a positive skin prick test to house dust mite (HDM; D. farinae and/or D. pteronyssimus), among which 41 (40.1%) were also positive to grass pollen (n = 29) and animals (n = 11).”
It was mistake. We corrected to: All 106 children (100%) had a positive skin prick test to house dust mite (HDM; D. farinae and/or D. pteronyssimus), among which 40 (37.7%) were also positive to grass pollen (n = 29) and animals (n = 11).
- Table 1. Remove the line with the “106 (100%)”, this can be included as a footnote.
We have done
- Table 1. Change the name to something more accurately such as “Clinical and lung function characteristics of patients.”
We changed to: “Clinical and lung function characteristics of patients.”
- Line 166, the title is eating habits but talks also about physical activity, so include it in the title.
We changed to: Eating habits and physical activity
- Line 172 and 193. “Children with asthma” were defined as AA, so use the definition.
Yes, we have changed to : Children with AA ate more frequently than children with AR (χ² = 12.9; p = 0.04) and Children with AA ate the last meal 1 hour before sleep more frequently than those in the AR group.
- Line 192. “from 2.5 to 6 hours before bedtime”, please check this sentence, it is strange to think that last meal was 6 hours before sleep.
It was mistake , should be : …19.8% (n = 21) did so much earlier (from 2.5 to 3 hours before bedtime)…
- Line 197, the definition PE has not been defined so write both words.
We have done PE- physical education
- Line 203. (same as before) in the title add that not only nutritional but also body composition will be shown.
We added information. Nutritional status and body composition
- Line 205. Please review, I think this is incorrect: “obesity was present in 11.1% compared to 6.9% in the AR group”, while according to the table, obesity was 4.7 and 6.9%.
Yes, it was mistake. We have changed to: Obesity was diagnosed in six children (6.0%) and eight were overweight (7.5%). In the AA group, obesity was present in 4.7% compared to 6.9% in the AR group
- Line 211. All the manuscript has been divided into small sections, and here the multifactorial linear regression analysis was just introduced. Please add a title.
We added title – The multifactorial linear regression analysis
- Line 217. Add to the figure 2 caption, the meaning of yellow, red, grey and black lines.
We added information Legend: red line means < 25th percentile, black line 25th-89th percentile, grey line 90th-97th percentile and yellow line >97th percentile
- Table 4, please add in the footnote the meaning of B and Beta
We added information in the footnote
- Line 245, this is a formal text so I recommend you to avoid contractions such as “obesity’s”.
Yes, we have changed to: The cause of impact of the obesity on asthma risk is still unknown
- Lines 327-330. This paragraph was fully copied from results. Avoid to duplicate the information, this will make to the reader to lose attention or credibility to the work. So eliminate or re-write in other form to introduce the discussion.
We decided changed fragment to: In our study, children from both groups (AR and AA) in over 50% reported low physical activity. It is well known that a lack of exercise increases the risk of obesity.
- Lines 331 to 334. This sentence is not understood. What are the 2 groups compared in the ISAAC trail phase 3.
We corrected to: . This was confirmed by research in the Phase 3 ISAAC trial, in which television viewing (5+ hours/day vs. <1 hour/day, p < 0.001) (group with low physical activity) was statistically significant associated with higher BMI in comparison to vigorous physical activity (3+ hours/week vs. never, p < 0.001) (group with high physical activity) in adelescents.
- Line 335. “current behaviors are more important than other factors”, instead of write other factors, name them.
We changed to: authors also suggested that current behaviors are more important than other factors such as birth weight, breastfeeding, , current maternal or paternal smoking in the early childhood in development of obesity
- Line 346. “The value of our study is its complex examination…”, I do not understand what the authors mean with this sentence, but I suggest to use any idea from the begging of my letter.
We have changed to: Our study provides two very important issues in the study of respiratory affections such as allergy and asthma. These are the fact that this is the first study on nutrition carried out in newly diagnosed AR teenagers, before any medication that could mislead any result, and the fact that this is another study from few existent about nutrition in respiratory allergy.
- Finally, sorry to be so insistent in this but for the scientific community, is important that p value, is in italics as p.
We corrected throughout the manuscript

Reviewer 3 Report
The reviewer comments have been addressed but the authors should be aware that in future studies, the findings would be dramatically strengthened with incorporation of a control group.
Author Response
Thank you for accepting our correction.
We agree with the Reviewer sugestions and plan to examine the control group in our further research
This manuscript is a resubmission of an earlier submission. The following is a list of the peer review reports and author responses from that submission.
Round 1
Reviewer 1 Report
It is an interesting paper about the eating habits of children with allergic rhinitis compared to children with both allergic rhinitis and asthma.
I have major concerns :
- It is clear that dieatary habits may have an important role in the development of allergic diseases, however in this study I do not think that we can demonstrate this role. Indeed, we cannont know if their habits have appeared after they were diagnosed with an allergic disease or whether they were already having these habits before they were sisick.
- It is quite obvious that a high BMI is linked to bad eating habits and low physical activities. The results found do not bring up anything original.
- Nothing have been mentionned about the reasons why these children have a low/moderate physical activities.
- The paper should be reviewed by an english speaking person.
Reviewer 2 Report
To the authors:
The manuscript entitled “Allergic children – a new target to correct dietary habits.” By Eliza Wasilewska, Sylwia Małgorzewicz, Marta Gruchała-Niedoszytko, Magdalena Skotnicka, Ewa Jassem, describes the study of the eating habits and risk factors of obesity and pulmonary functions in 106 children with allergic rhinitis alone or with asthma.
My general comments are:
The aim of manuscript is very interesting and novel as I looked into the available publications and the topic has not been described before, however it has significant flaws that should be addressed by the authors.
Certainly, allergic diseases are arisen worldwide and especially in children and teenagers. However, there are two main types of allergy in children, the food allergy and the respiratory allergy. In this case, the manuscript focuses in the second one, although most literature about nutrition status in allergic children relays on food allergy. Therefore, I suggest changing the title of the manuscript accordingly to be more consistent to the content. Moreover, the sample size is small and the patients are from a specific place, so both characteristics should be included in the title.
The second general comment is focused to the discussion part, as this seems not being supported by the results. Obesity and overweight in teenagers are in range of healthy polish population, however, according to the authors these conditions increase risks associated with inflammation, immune system activation, asthma development and an increase of clinical symptoms. However, none of these aspects have been analyzed by the authors and it can be these conditions are only the outcome to gain weight in adolescence. Therefore, please check the discussion and make it clear that obesity and overweight are important conditions that might impact later during the development of the disease but that in the case of the patients analyzed, this can not be concluded and that prospective studies are needed. I recommend the authors to write the discussion focused on your findings and how they can match with the facts from the literature.
Another example, line 233 “had eaten later their last meal before sleeping (χ² =19.4; p=0.001).” , this is not entirely true as it was significant only for those children with had their last meal 1 hour before sleeping.
The third general comment is in relation to the methods description and the clinical data of the patients. Please, I think it is missing a table with the results of the ISAC, the age of onset of allergic rhinitis, gender, family history of allergies. In addition, please add to M&M, on the description of the asthmatic patients these were newly asthma diagnosed. Add to the table medication for allergy treatment.
Minor comments:
- 1 is not fully understandable, please explain in the main text of the manuscript or adapt the figure.
- Fig 1. Shows the “QQ5” questionnaire but later you named as 6-FFQ (Food Frequency Questionnaire).
- Line 107. Please explain briefly a description of the 6-FFQ Questionnaire, and describe if it was used crude or standardized, and explain the meaning of the 6. Check for reference 19, I did and does not really explain the 6-FFQ questionnaire.
- Please revise those allergens that must be in italics such as Dermatophagoides farinae.
- Line 122. “on OLAF /OLA centile”, please define this parameter as in internet appear in polish language.
- Lines 128 to 132 can be reduced to a single sentence: Differences for somatic traits and, between AR and AA were evaluated using Student's t-test or, for asymmetrical distributions
- Line 130 “AB” is incorrect, it is “AA”, please revise throughout the text.
- Line 176. “(χ² =19.4; df=5, p=0.005).” add or delete df value to all the chi-square tests.
- Results in lines 166-167 are not included in a table. Maybe another table is needed.
- Line 125, “bioimpedence method (BIA)” was defined but later BIA was never used.
- Line 293, “Children with allergic rhinitis present incorrect eating habits;”, please define which are these habits.
- Lines 293-294, “risk factor for obesity and then asthma development.” Please add the reference that obesity goes first to asthma development.
- Please check the manuscript entirely as the sample size is not so high and with many sub-classes the statistical power is poorer. In those causes, it is better to talk as the results “suggest something or points to something”
- Finally, all p values, p must be in italics.
Reviewer 3 Report
The manuscript titled “Allergic children – a new target to correct dietary habits” details a clinical study which compared the eating habits, body parameters, and exercise to the diagnosis of allergic rhinitis (AR) and rhinitis plus asthma (AA). There are some concerns that need to be addressed prior to publication.
- One of the most concerning issues with the manuscript is that there is no control group. Comparisons to an equivalent group of children without AR might provide substantial differences and has the potential to provide new understandings about AR.
- The age range for the children is quite broad. There was no explanation for the choice of age; potentially there might other insights if the children were divided into at least 2 age groups.
- The English grammar and usage of words is incorrect in many cases. The manuscript should be proofread and corrected by an English speaker.
